# Vitamin D Receptor Genetic Variation and Cancer Biomarkers among Breast Cancer Patients Supplemented with Vitamin D3: A Single-Arm Non-Randomized Before and After Trial

**DOI:** 10.3390/nu11061264

**Published:** 2019-06-04

**Authors:** Elham Kazemian, Mohammad Esmaeil Akbari, Nariman Moradi, Safoora Gharibzadeh, Alison M. Mondul, Yasaman Jamshidi-Naeini, Maryam Khademolmele, Katie R. Zarins, Nasim Ghodoosi, Atieh Amouzegar, Sayed Hossein Davoodi, Laura S. Rozek

**Affiliations:** 1Department of Basic Sciences and Cellular and Molecular Nutrition, Faculty of Nutrition Sciences and Food Technology and National Nutrition and Food Technology Research Institute, Shahid Beheshti University of Medical Sciences, No 7, Hafezi St. Farahzadi Blv, Shahrake Gharb, Tehran 19816-19573, Iran; kazemianelham@yahoo.com; 2Endocrine Research Center, Research Institute for Endocrine Sciences, Shahid Beheshti University of Medical Sciences, Tehran 19395-4763, Iran; amouzegar@endocrine.ac.ir; 3Cancer Research Center, Shahid Beheshti University of Medical Sciences, Tehran 19899–34148, Iran; profmeakbari@gmail.com; 4Department of Clinical Biochemistry, Faculty of Medicine, Kurdistan University of Medical Sciences, Sanandaj 66177-13446, Iran; nariman8463@yahoo.com; 5Endocrine Research Center, Institute of Endocrinology and Metabolism, Iran University of Medical Sciences, Tehran 14496-14535, Iran; 6Department of Epidemiology and Biostatistics, Pasteur Institute of Iran, Tehran 13169-43551, Iran; safoora.gharibzadeh@gmail.com; 7Department of Epidemiology, University of Michigan School of Public Health, Ann Arbor, MI 48109-2029, USA; amondul@umich.edu; 8Department of Nutritional Sciences, Texas Tech University, Lubbock, TX 79409, USA; y.jamshidinaeini@ttu.edu; 9Department of Nutrition Science, Faculty of Medical Science and Technology, Islamic Azad University, Science and Research Branch (SRBIAU), Tehran 14778-93855, Iran; maryam@khademolmele.ir; 10Department of Environmental Health Sciences, University of Michigan School of Public Health, M6529 SPH II, 1415 Washington Heights, Ann Arbor, MI 48109-2029, USA; kmrents@umich.edu; 11Department of Community Nutrition, School of Nutritional Sciences and Dietetic, Tehran University of Medical Sciences, Tehran 14155-6446, Iran; Nasim_ghodoosi@yahoo.com

**Keywords:** vitamin D3, plasma 25(OH)D, breast cancer survivor, vitamin D receptor, nutrigenomics

## Abstract

We investigated whether vitamin D receptor (*VDR*) polymorphisms were associated with cancer biomarkers, i.e., E-cadherin, matrix metallopeptidase 9 (MMP9), interferon β (IFNβ), soluble intercellular adhesion molecule-1 (s-ICAM-1), soluble vascular cell adhesion molecule-1 (s-VCAM-1), tumor necrosis factorα (TNFα), interleukin 6 (IL6), plasminogen activator inhibitor-1(PAI-1), and human high sensitivity C-reactive protein (hs-CRP), among breast cancer survivors who received vitamin D3 supplementation. In a single-arm non-randomized pre- and post trial, 176 breast cancer survivors who had completed treatment protocol including surgery, radio and chemotherapy were enrolled in the study and received 4000 IU of vitamin D3 daily for 12 weeks. The association between the *VDR* SNPs (*ApaI*, *TaqI*, *FokI*, *BsmI* and *Cdx2*) and response variable changes was assessed using linear regression, utilizing the “association” function in the R package “SNPassoc”. We observed that women with AA and GA [codominant model (AA compared to GG) and (GA compared to GG); dominant model (AA & GA compared to GG)] genotypes of *Cdx2* showed higher increase in plasma MMP9 levels compared to the GG category. In addition, carriers of *BsmI* bb showed greater decrease in circulating TNFα levels after vitamin D3 supplementation [recessive model (bb compared to BB & Bb]. Likewise, significant associations were identified between haplotypes of *VDR* polymorphisms and on-study plasma MMP9 changes. However, our results indicate that *VDR* genetic polymorphisms were not associated with longitudinal changes in the remaining cancer biomarkers. Overall, our findings suggest that changes in certain inflammatory biomarkers in breast cancer survivors with low plasma 25(OH)D levels, supplemented with vitamin D3, may depend on *VDR* SNPs and haplotypes.

## 1. Introduction

Breast cancer is the most common cancer among women in both the developed and developing countries and is the leading cause of death from cancer among women worldwide [1]. Importantly, due to increased surveillance and improved treatment, the breast cancer survivorship rate is steadily rising [2]. Breast cancer survivors have complex health concerns, related to surgery, radiation therapy, chemotherapy, and hormonal therapy; which include cardiovascular disease, increased risk of second malignancies, bone health, fatigue, weight management, pain and menopausal syndrome [2]. The identification of inexpensive and easy to implement strategies to improve survivorship will be crucial in low- and middle income countries where the the burden is the greatest and breast cancer incidence is increasing rapidly.

Vitamin D3 is hypothesized to have many beneficial health effects [3]. In addition to the potential anti-carcinogenesis effects of vitamin D [4,5,6,7,8,9,10,11,12,13,14,15,16,17,18], higher circulating levels of 25-hydroxy vitamin D (25(OH)D) have been associated with better overall survival, breast cancer specific survival (BCSS), relapse free survival (RFS) and invasive disease free survival (IDFS) in women [8,19]. Preclinical studies suggest that 1,25-Dihydroxyvitamin D3, calcitriol, (1,25(OH)2D3) mediates anti-carcinogenesis effects through suppression of inflammation, extracellular proteases, adhesion molecules, down regulation of the NF-κβ signaling and enhanced expression of the tumor suppressor gene E-cadherin [4].

The vitamin D receptor (*VDR*) is a crucial mediator for vitamin D genomic actions including regulating transcription of several genes involved in cellular growth, differentiation, apoptosis, angiogenesis, inflammation and metastasis [19,20]. The complex nature of the *VDR* gene regulation and a large number of *VDR* binding sites recognized in genome wide studies suggest contribution of variations in the *VDR* gene in many processes involved in carcinogenesis and overall health [20,21]. Results of previous studies indicate that variations in *VDR* are associated with cancer incidence, mortality and survival [8].

Despite extensive research on the roles of vitamin D or *VDR* per se in carcinogenesis [1,2,3,4,5,6,7,8,9,10,11,12,13,14,15,16,18,19,20] to date, no controlled trial has been performed to investigate the possible biological interactions of the *VDR* genetic variations with vitamin D intake on diverse aspects of the breast cancer host response. Additionally, results of studies assessing the association of vitamin D and its receptor independently with cancer incidence and prognosis are conflicting [8,22,23]. Findings from experimental studies of vitamin D3 intervention on associated primary 1,25(OH)2D3 target gene expression suggest heterogeneous inter-individual responses to vitamin D3 supplementation [24]. Likewise, *VDR* expression and function including receptor affinity, binding to nuclear DNA and RNA transcription may be influenced by variation in the *VDR* gene [25]. We hypothesize that the biological interactions between genetic variation in the vitamin D pathway gene and vitamin D status may play a crucial role in cancer development, prognosis and contribute to discrepancies in findings of previous studies. Indeed, nutrigenomic approaches, which use gene polymorphism data to estimate nutrient transportation, metabolization and requirements to predict the efficiency of nutrient uptake, were not explored in previous studies conducted on cancer and vitamin D. Future trials using genetic and genomic testing along with vitamin D levels can classify risk of vitamin D deficiency and facilitate personalized primary cancer prevention and treatment. To address this gap in our literature, here for the first time, we investigated whether *VDR* polymorphisms (*ApaI*, *TaqI*, *FokI*, *BsmI* and *Cdx2*) could affect breast cancer survivors’ responses to vitamin D3 supplementation through potential cancer biomarkers, including E-cadherin, matrix metallopeptidase 9 (MMP9), interferon β (IFNβ), soluble intercellular adhesion molecule-1 (s-ICAM-1), soluble vascular cell adhesion molecule-1 (s-VCAM-1), tumor necrosis factorα (TNFα), interleukin 6 (IL6), plasminogen activator inhibitor-1 (PAI-1), and human high sensitivity C-reactive protein (hs-CRP). 

## 2. Materials and Methods

### 2.1. Study Participants

Considering the high survival rate, breast cancer survivors were selected as a target population for our investigation [26]. The detailed study protocol was described elsewhere [27]. In brief, This study was conducted in breast cancer survivors who were admitted to Shohaday-e-Tajrish hospital and its associated clinics in Tehran for cancer follow up. Patients who were diagnosed with breast cancer at least six months before the study enrollment and had completed treatment (including surgery, radio and chemotherapy) were eligible to participate in the study. Two hundred and fourteen subjects were recruited (Appendix A). Eligibility criteria were: Age 25–65 years; body mass index (BMI) between 18.5 to 35 kg/m^2^; no use of vitamin D3 supplements (1000 IU daily or 50,000 IU weekly or 300,000 intramuscular injections) for at least four months before entry to the study; no use of dietary and herbal supplements during the intervention period. Exclusions included: History of malabsorption syndrome; calcium metabolism disorders; gastrointestinal, renal, inflammatory and other endocrinological diseases which might interfere with the study; undergoing treatment for weight reduction; plasma 25(OH)D ≥ 100 nmol/L; pregnancy.

### 2.2. Study Design and Intervention

Eligible participants received 4000 IU of vitamin D3 daily for 12 weeks in a single-arm non-randomized pre- and post trial (cholecalciferol tablets manufactured by JALINOUS PHARMACEUTICAL CO., Tehran, Iran). Without increasing the risk of overdose, health related benefits of vitamin D were observed by circulating 25(OH)D levels of 75 to 110 nmol/L obtained through daily intakes of vitamin D3 in the range of 1800 to 4000 IU [28]. The intervention period was limited to winter and spring months to minimize cutaneous synthesis of vitamin D3. Biweekly telephone calls involving a five to eight min assessments were made to minimize non-compliance. To assess intervention compliance of participants, they were asked to return their empty bottle of pills at the end of the study. Adherence was calculated by the number of pills consumed divided by the numbers expected to be taken during the study and presented as a percentage.

All participants read and signed the informed consent form at the beginning of the study. This trial has been registered in the Iranian Registry of Clinical Trials (IRCT) under the identification code: IRCT2017091736244N1, registration date: 10-11-2017, http://www.irct.ir/trial/27153.

### 2.3. Study Measurements

Demographic, background and pathologic data were obtained through a questionnaire and review of medical records at recruitment. Anthropometric measurement, dietary intakes, sun exposure, physical activity and laboratory assessments were conducted at enrollment and after 12 weeks of intervention.

### 2.4. Anthropometrics and Dietary Measurements

Height was measured using the Secastadiometer with subjects looking ahead without footwear. Body weight was measured without shoes in light indoor clothing, using a digital scale to the nearest of 0.1 kg. Body mass index was calculated by dividing weight in kilogram by the square of height in meter. Waist circumference (WC) was measured at the midpoint between the highest point of the iliac crest and the last floating rib. Hip circumference (HC) was measured by placing measuring tape at the maximal circumference over the buttocks. A twenty-four-hour food record for three days including a weekend day and two working days was applied to estimate average dietary intakes.

### 2.5. Physical Activity Assessment

Physical activity was assessed by the International Physical Activity Questionnaire (IPAQ) comprising seven questions translated by the Iran’s National Elites Foundation [29]. Total physical activity was reported as Metabolic Equivalent Task minutes per week (MET-min/week) and was grouped as light PA (MET < 600 min/week), moderate PA (600 ≤ MET < 1500 min/week) and vigorous intensity PA (MET ≥ 1500 min/week) [30].

### 2.6. Sun Exposure Assessment

Sun exposure was assessed by a questionnaire developed to measure sun-related behavior [31]. Subjects were requested to report hours of outdoor activities over the previous week and body surface area (BSA) exposed to sunlight while outdoors. A sun exposure index was calculated by multiplying the percentage of BSA exposed by the hours of exposure to sunlight per week [31].

### 2.7. Laboratory Measurements

Participants’ fasting venous blood samples were taken at the beginning and end of the study. All laboratory measurements were carried out at the laboratory of the Department of Biochemistry, Faculty of Medicine, Iran University of Medical Sciences.

### 2.8. DNA Extraction and Genotyping

Genomic DNA was extracted from white blood cells (WBC), using the Gene All 100 DNA Blood Kit (Gene All Biotechnology Co., Ltd., Seoul, Korea). The ratio of the absorbance at 260 and 280 nm was used to determine DNA concentration and yield. Polymerase chain reaction (PCR) amplification was performed to amplify specific DNA target. Restriction fragment length polymorphism (RFLP) was used for *VDR* genotyping at *ApaI*, *TaqI*, *FokI* and *BsmI* SNPs and theTetra amplification-refractory mutation system (ARMS) method for *Cdx2* explained elsewhere [27]. 

### 2.9. Plasma 25(OH)D Measurement

We assessed plasma 25(OH)D at the beginning and after 12 weeks of supplementation using the ELISA kit (Euroimmun, Lübeck, Germany) according to the manufacturer’s protocol. Intra- and inter-assay coefficient of variations (CVs) were 5% and 7.8%, respectively. 

### 2.10. Inflammatory, Proliferative and Differentiative Biomarkers Measurement

Preclinical studies have suggested that 1,25(OH)2D mediates anti-carcinogenesis effects through suppression of inflammation, extracellular proteases, the tumor suppressor gene E-cadherin, adhesion molecules and down regulation of the NF-κβ signaling [4]; thus, biomarkers of proposed pathways, including E-cadherin, matrix metallopeptidase 9 (MMP9), interferon β (IFNβ), soluble intercellular adhesion molecule-1 (s-ICAM-1), soluble vascular cell adhesion molecule-1 (s-VCAM-1), tumor necrosis factorα (TNFα), interleukin 6 (IL6), plasminogen activator inhibitor-1(PAI-1), and human high sensitivity C-reactive protein (hs-CRP) were chosen as outcomes of interest.

Plasma inflammatory biomarkers were assessed by determining plasma interleukin 6 (IL6) (Invitrogen, Carlsbad, CA, USA), tumor necrosis factorα (TNFα) (Invitrogen, Carlsbad, CA, USA), human high sensitivity C-reactive protein (hs-CRP) (Bio Vendor, Brno, Czech Republic) and interferon β (IFNβ) (R&D Systems, Inc., Minneapolis, MN, USA) levels using the ELISA kit. 

ELISA was used to determine plasma plasminogen activator inhibitor-1(PAI-1) (EBIOSCIENCE, Vienna, Austria), E-cadherin (R&D Systems), matrix metallopeptidase 9 (MMP9) (EBIOSCIENCE, Vienna, Austria), soluble intercellular adhesion molecule-1 (s-ICAM-1) (BioVendor, Brno, Czech Republic), and soluble vascular cell adhesion molecule-1 (s-VCAM-1) (BioVendor, Brno, Czech Republic), as proliferation, differentiation and metastasis measures.

### 2.11. Statistical Analysis

The Shapiro-Wilk test was used to test the normality of distribution of continuous variables. The difference between two skewed continuous variables was compared using the Wilcoxon signed-rank test. Vitamin D receptor *ApaI*, *TaqI*, *FokI*, *BsmI* and *Cdx2* polymorphisms were modeled as the main exposures. On-study variables changes were calculated by subtracting the value of pre- from the value of post intervention and non-normal variables were log-transformed. The association between the selected *VDR* SNPs and response variables changes (E-cadherin, MMP9, IFNβ, s-ICAM-1, s-VCAM-1, TNFα, IL6, PAI-1 and hs-CRP ) in breast cancer survivors supplemented with vitamin D3 was assessed using linear regression, performing the “association” function in the R package “SNPassoc” [32,33]. Then, the regression coefficient (β) with 95% confidence interval (CI) and *p* value was reported. An association analysis between *VDR* SNPs and response variables was carried out under five different genetic models including codominant, dominant, recessive, over dominant and log-additive. One of the main assumptions for the linear regression analyses is the homogeneity of variance of the residuals. To test this assumption, we used the Breusch–Pagan test. In the case of heteroscedasticity, we re-built the model with *y*-transformation such as the Box–Cox transformation.

Factors that were significantly associated with changes in response variables in our study, or known or hypothesized to be associated with breast cancer were considered as potential confounders (age, dietary intakes of calcium, vitamin D, energy, fat, protein and carbohydrate, use of calcium and vitamin D supplements at baseline, calcium supplement use during the study, sun exposure, BMI, WC, HC, oral contraceptive pill (OCP) use, parity, menopausal status, breast cancer staging, baseline 25(OH)D concentration, and on-study 25(OH)D changes). Factors were included if their addition to the model changed the main effect by 10 percent. The final multivariable model has been adjusted for age, BMI and 25(OH)D levels at the baseline. Regression analyses were conducted on 176 participants who took ≥90% of their pills, with no gaps >5 days and no personal vitamin D supplementation during the study. We used false-discovery rate (FDR) methods to correct for multiple comparisons [34].

Causal mediation analysis (CMA) was conducted to determine to what extent, the total effect of the *VDR* (as a main exposure) is mediated by changes in plasma 25(OH)D (as an intermediate variable) on the causal pathway between the main exposure and the outcome of interest [35].

Distributions from the Hardy–Weinberg equilibrium (HWE) for each SNP were assessed using an exact test, performed by the HWE exact function in the R package “HardyWeinberg”, with *p* values < 0.05 being considered significant. R package ‘haplo.stats’ was employed to test the association of estimated haplotypes with changes in response variables [36]. The Haplo score was adjusted for age, BMI and 25(OH)D levels at the baseline. All *p* values were presented for a two-tailed test and *p* < 0.05 was considered statistically significant. Statistical analyses were performed using Stata 14.0 (StataCorp., College Station, TX, USA, 2015) and the computing environment R Version 3.4.3 (R Development Core Team, 2017. R: A language and environment for statistical computing. R Foundation for Statistical Computing, Vienna, Austria. URL https://www.R-project.org/).

## 3. Results

Two hundred and fourteen women previously diagnosed with breast cancer at the Shohadaye Tajrish Hospital, Tehran were enrolled in the study. The final analysis was carried out on 176 women, after excluding subjects who did not take >10% of their study pills (*n* = 14) and individuals lost to follow up (*n* = 24) (Appendix A). All genotype distributions were in the Hardy–Weinberg equilibrium proportions. The average survival time for study subjects from diagnosis to recruitment was 4.5 years (range 1–16).

The mean age of the study participants was 49.0 (SD = 8.8) years, 78 percent of whom were married and only four (2%) subjects were smokers. Ninety-five women (53.9%) were diagnosed with ER^+^PR^+^ breast cancer and twenty (11.3%) were diagnosed with ER^−^PR^−^HER^2−^ breast cancer. Thirty (17%) and 127 (72%) of study participants had a family history of breast cancer and were menopausal, respectively. Thirty-four (23%), 75 (44%) and 42 (27%) of study patients were diagnosed with stages I, II and III breast cancer, respectively. Ninety-two (52%) and 49 (28%) had low or moderate intensity physical activity, and the median energy intake of all participants was 1906 Kcal/day (IQR, 1835–2046) (Table 1).

Comparison of changes in response variables before and after vitamin D3 supplementation are displayed in Table 2. Despite biomarkers’ changes data not being normally distributed, due to the sufficiently large sample size, using non-parametric methods led to the same results as using methods that rely on mean and standard deviation (SD). Baseline 25(OH)D levels ranged from 2.49 to 180.21 nmol/L (mean, 41.5; SD, 27.5) and reached an average of 113.1 (SD, 45.8) at the end of the study after supplementation; the mean post-supplemental change in plasma 25(OH)D was 71.5 nmol/L (range –7.98 to 200.4). Considering the recommended level of ≥75 nmol/L, 159 (87%) women were vitamin D deficient or insufficient at the baseline whereas 141 (74%) of them had desirable levels of plasma 25(OH)D at the end of the study after vitamin D3 treatment. We found significant intra-individuals variations in response variables before and after treatment with vitamin D3 including: Plasma 25(OH)D (mean pre-treatment, 41.5 compared to mean post-treatment, 113.1), E-cadherin (mean pre-treatment, 120.4 compared to mean post-treatment, 112.8), MMP9 (mean pre-treatment, 1746 compared to mean post-treatment, 1655), s-ICAM-1(mean pre-treatment, 672.5 compared to mean post-treatment, 650.8), TNFα (mean pre-treatment, 20.6 compared to mean post-treatment, 19.3), PAI-1 (mean pre-treatment, 46.2 compared to mean post-treatment, 44.4) and hs-CRP (mean pre-treatment, 2.69 compared to mean post-treatment, 2.55) (Appendix A).

With respect to changes in outcomes by the genetic variation in *VDR*, our adjusted analyses indicate women with tt genotypes of *TaqI* [codominant model (tt compared to TT): −0.58 (−0.95, −0.20); and the recessive model (tt compared to TT & Tt): −0.57 (−0.92, −0.21)] experienced higher decrease in plasma MMP9 levels compared to TT. However, only the observed association under the recessive model remained significant after FDR correction (Figure 1b).

Additionally, individuals with the AA and GA genotypes of *Cdx2* [codominant model (AA compared to GG): 0.60 (0.21, 0.99) and (GA compared to GG): 0.45 (0.23, 0.68); dominant model (AA & GA compared to GG): 0.48 (0.27, 0.69)] were found to have a greater increase in plasma MMP9 levels compared to the GG category which remained statistically significant even after correction for multiple testing (Figure 1a). Furthermore, carriers of *BsmI* bb showed a larger decrease in circulating TNFα levels after vitamin D3 supplementation [recessive model (bb compared to BB & Bb): −0.24(−0.39, −0.08)], the latter association remaining statistically significant after FDR correction. (Table 3 and Table 4, Figure 1c).

Haplo.score analyses were performed by considering breast cancer biomarker changes as a quantitative trait (Table 5). At first, we evaluated a combination of all SNPs (five SNPs) determined in the current study as one haplotype and then, based on these results as well as preliminary findings of other studies [37,38,39], we limited the haplotype to three-SNP haplotypes: *BsmI ApaI TaqI*; *FokI BsmI ApaI*; *Cdx2 FokI BsmI*; and *FokI TaqI Cdx2*. Haplotype score analyses indicated that response to vitamin D3 supplementation was mediated by haplotypes of *VDR* (Table 5). We identified significant associations of plasma MMP9 changes in response to vitamin D3 supplementation with haplotypes of *Cdx2 FokI BsmI ApaI TaqI* (*p* value < 0.001)*, Cdx2 FokI BsmI* (*p* value = 0.003) and *FokI TaqI Cdx2* (*p* value < 0.001) after multiple testing correction. Furthermore, we found a significant association between circulating plasma s-ICAM-1 changes and *Cdx2 FokI BsmI ApaI TaqI* (*p* value = 0.03)*, Cdx2 FokI BsmI* (*p* value = 0.02) and *FokI TaqI Cdx2* (*p* value = 0.03) haplotypes which did not remain statistically significant after correction for multiple testing (Table 5). As presented in Appendix A, inverse associations of circulating MMP9 changes in response to vitamin D3 treatment with GfbAt (*Cdx2* G, *FokI* f, *BsmI* b, *ApaI* A, *TaqI* t, *p* value = 0.002) and GFbaT(*Cdx2* G, *FokI* F, *BsmI* b, *ApaI* a, *TaqI* T, *p* value = 0.03) haplotypes and positive associations with AFBAT(*Cdx2* A, *FokI* F, *BsmI* B, *ApaI* A, *TaqI* T, *p* value < 0.001), AFBAt (*Cdx2* A, *FokI* F, *BsmI* B, *ApaI* A, *TaqI* t, *p* value < 0.001), AfbAT (*Cdx2* A, *FokI* f, *BsmI* b, *ApaI* A, *TaqI* T, *p* value = 0.03) and AFBaT (*Cdx2* A, *FokI* F, *BsmI* B, *ApaI* a, *TaqI* T, *p* value = 0.03) was shown. Likewise, Gfb (*Cdx2* G, *FokI* f, *BsmI* b, *p* value < 0.001), GFb (*Cdx2* G, *FokI* F, *BsmI* b, *p* value = 0.02), fTG (*FokI* f, *BsmI* T, *Cdx2* G, *p* value = 0.005), ftG (*FokI* f, *TaqI* t, *Cdx2* G, *p* value = 0.007) and FTG (*FokI* F, *TaqI* T, *Cdx2* G, *p* value = 0.03) haplotypes were associated with a significant decrease in plasma MMP9 over the study while AFB (*Cdx2* A, *FokI* F, *BsmI* B, *p* value < 0.0001), FTA (*FokI* F, *TaqI* T, *Cdx2* A, *p* value < 0.0001), FtA (*FokI* F, *TaqI* t, *Cdx2* A, *p* value < 0.0001) and fTA (*FokI* f, *TaqI* T, *Cdx2* A, *p* value for Haplo.Score = 0.01) were associated with an increase in circulating MMP9 levels (Appendix A).

The inclusion of plasma 25(OH)D changes as a mediator variable in causal mediation analysis revealed that neither the association of *TaqI* and *Cdx2* with MMP9 nor the *BsmI* with TNFα were mediated by plasma 25(OH)D changes during the study (Table 6).

## 4. Discussion

In these analyses, we found possible associations between changes in plasma MMP9 and *VDR Cdx2* and *TaqI* SNPs, and associations were also found between TNFα changes and the *VDR BsmI* polymorphism, even after FDR correction. Moreover, we found that inferred haplotypes from the *ApaI, TaqI, FokI*, *BsmI* and *Cdx2* polymorphisms were significantly associated with plasma MMP9 changes after 12-weeks supplementation with vitamin D3. However, our findings showed that *VDR* genetic polymorphisms were not associated with longitudinal changes in the remaining cancer biomarkers among breast cancer survivors, who received vitamin D3 supplementation.

Preclinical and experimental studies indicate that 1,25(OH)2D mediates anti-carcinogenesis effects through multiple pathways via *VDR*, a crucial mediator for the vitamin D genomic actions: (1) Suppression of extracellular proteases, including MMP9, urokinase-type plasminogen activator, tissue type plasminogen activator, and adhesion molecules (ICAM and VCAM) resulting in inhibition of cancer invasion and metastasis; (2) expression of the tumor suppressor gene E-cadherin induced by calcitriol results in reduced metastatic potential, partly through the promotion of E-cadherin-mediated cell-cell adhesion; (3) down regulation of NF-κβ signaling, one of the key players of inflammation and cancer and (4) suppression of anti-apoptotic genes such as *BCL-2* and *BCL-XL* and also increasing expression of pro-apoptotic genes including *BAX*, *BAK* and *BAD* [4]. Study outcomes were selected based on potential roles of calcitriol in these pathways. 

To the best of our knowledge, most results of existing studies are not directly comparable with our work; there are however a few observational studies concerning biological interactions of vitamin D intake and polymorphisms in the vitamin D pathway gene and risk of cancer [40,41,42,43]. For example, a case-control study of Korean participants by Eom et al. indicates that the effect of vitamin D on gastric carcinogenesis was not modified by the genetic polymorphisms of vitamin D-related genes [44]. In another case-control study of a US population, the interaction between SNPs or haplotypes within the *VDR* gene and dietary factors did not affect the risk of colorectal cancer [40]. Lowe et al. report that low levels of circulating 25(OH)D (<50 nmol/L) in combination with bb *BsmI VDR* genotype was accompanied by an increased risk of breast cancer in the UK Caucasian population [41]. In a study by Takeshige et al., *FokI* polymorphism modulated the effect of vitamin D intakes on the risk of colon cancer while no interaction was found between *BmsI*, *TaqI*, *ApaI* SNPs and dietary vitamin D intakes with colorectal, colon, or rectal cancer risk [42]. Moreover, two randomized trials examined how genetic variation in the *VDR* affects individuals’ responses to vitamin D3 supplementation. In a randomized double blind control trial of patients with Parkinson disease (PD), the effect of vitamin D3 (1200 IU/day for 12 months) to stabilize PD were only associated with the *VDR FokI* TT or CT but not the other *SNPs* examined (i.e., *BsmI*, *Cdx2*, *ApaI* and *TaqI)* [43]. Another study conducted among diabetic patients, consuming 500 mL yogurt drink (doogh) fortified with 1000 IU vitamin D3 for 12 weeks reported that individuals with the *VDR FokI* ff genotype had lower response to vitamin D3 intake in terms of circulating 25(OH)D, serum hsCRP and IL6, while no association was found between vitamin D3 intake and the *VDR FokI* in relation to serum MMP-9, TNFα and IFN_γ_ changes, results inconsistent with those of our study. However, it is noteworthy to mention that hsCRP and IL6 had a large standard deviation in the ff genotype group indicating that the observed association could not be clinically significant because of a large variation in study measures or presence of outliers [45]. Moreover, the diseases and vitamin D3 dosage were different in these studies, and hence not comparable with the findings of the current study [46]. In the Cavalcante et al. study of elderly subjects, circulating 25(OH)D, parathyroid hormone (PTH), ultra-sensitive C-reactive protein (us-CRP), and alpha 1-acid glycoprotein (AGP-A) levels of those with the *VDR BsmI* BB/Bb were more responsive to vitamin D3 mega dose compared to the bb genotype [46]. Nevertheless, this study did not have enough power to elucidate the genotype–environment (GXE) interaction due to its small sample size. Moreover, the frequency distribution of the *VDR* SNPs varies among different populations, making it difficult to compare findings of different studies [47,48].

Moreover, we found that individuals with the haplotype containing *Cdx2* G *FokI* f *BsmI* b and *Cdx2* G *FokI F BsmI b* had the largest decreases in MMP9 in response to vitamin D3 supplementation. It is hypothesized that the haplotype based approach may be better than the one SNP at a time approach: (1) Properties and other features of resulting proteins depend on the amino acid combinations of polypeptide chains; (2) based on principles of population genetics, the individual’s variations are naturally determined by haplotype blocks; (3) lesser dimension of phased genotypes resulting in better statistical power of association tests [49].

To the best of our knowledge, there are only a few studies examining the association of haplotypes with the risk of breast cancer which were not directly comparable with our findings due to different haplotype definitions and outcomes examined. In a study by Abbas et al., the haplotype FtCA (*FokI* F, *TaqI* t, *VDR-5132* C, *Cdx2* A) was associated with increased risk of breast cancer compared to FTCG (the most prevalent haplotype) [39]. Since the G allele of *Cdx2* decreases the *VDR* gene transcription [50], haplotypes containing G allele are expected to be associated with increased risk of breast cancer; however G allele containing haplotypes including the FTCG haplotype in the Abbas study and GfbAt, Gfb and GfT in the current study were accompanied with decreased risk of breast cancer and better response to vitamin D3 supplementation, respectively [50,51,52]. Although reasons for these observations are not clear, it is noteworthy to mention that functional experiments indicated that other *VDR* primary promoter polymorphisms such as (G-1739A: rs11568820) and (A-1012G: rs4516035) rather than *Cdx2* may alter transcriptional activity of the *VDR* primary promoter [37]. In another prospective cohort study, the haplotype, GTCATTTCCTA (rs739837, rs731236, rs7975232, rs2239182, rs2107301, rs2239181, rs2238139, rs2189480, rs3782905, rs7974708, rs11168275) was identified to increase the risk of breast cancer by 50% [18]. McCullough et al. showed no association of breast cancer with haplotype containing *BsmI*, *ApaI*, *TaqI*, and a *poly(A)* which is consistent with findings of the current study [53]. In addition, with regards to the previously well-defined baT, BAt, and bAT haplotypes, findings of experimental studies are inconclusive, with some studies showing higher mRNA expression of the BAt haplotype compared to the baT, while others report the opposite pattern for *VDR* mRNA expression, stability and transactivation [37]. No significant association was noted between cancer biomarkers and the latter haplotype.

The current study, using nutrigenomic approaches, paid particular attention to genetic variation, diet and environment to develop personalized nutritional strategies to determine cancer risk factors. To the best of our knowledge this is the first trial which has examined the effects of biological interaction between *VDR* genetic polymorphisms and vitamin D intake on the diverse aspects of responses in breast cancer patients including inflammatory and immune biomarkers as well as those associated with cell proliferation, differentiation, damage and metastasis. The documented high prevalence of vitamin D deficiency and insufficiency among participants in the current study likely enhances the effects of vitamin D3 supplementation. Moreover, the vast array of variables measured allows us to discover the potential positive effects of vitamin D3. One limitation of our study was the small number of SNPs examined which most likely did not represent a large fraction of the variation in the *VDR*. In addition, the small sample size of the current investigation results in lower study power, particularly in terms of haplotype analyses, i.e., for rare haplotypes the number of individuals in each group was fairly low. Although we may lack sufficient power to precisely estimate the relationship, this does provides the utility of considering several variants when assessing the relationship between *VDR* and vitamin D related outcomes. The 12-week trial may not be long enough to observe anti-carcinogenesis effects of vitamin D3_._ Finally, the lack of a comparator group, as no randomization of the exposure (genetic variability) is possible, makes it very difficult to attribute any longitudinal changes in tumor markers to vitamin D3 per se, as these markers could have also improved over time without vitamin D3. Future well designed clinical trials study with larger sample sizes as well as a longer intervention time to correlate with long term complications of cancer (including overall survival) and different dosages of vitamin D are needed to verify our findings. In addition, future studies in cultured cells using the clustered regularly interspaced short palindromic repeats (CRISPR) [54] gene editing technology are needed to identify biological mechanisms for the functional effects of the *VDR* genetic variations mediating the effects of vitamin D3 supplementation on carcinogenesis.

## 5. Conclusions

In conclusion, our results indicate that changes in inflammatory biomarkers (plasma MMP9 and TNFα) associated with breast cancer risk and survival in breast cancer survivors with low plasma 25(OH)D levels, supplemented with vitamin D3 depends on *VDR* SNPs (*Cdx2*, *TaqI* and *BsmI*) and haplotypes. However, we found no significant association between *VDR* genetic polymorphisms and longitudinal changes in other cancer biomarkers measured in the current study. Those findings provide novel insights into a better understanding of which subsets of individuals are at greater risk of the adverse effects of vitamin D deficiency or those who may benefit most from normalization of the vitamin D status.

## Figures and Tables

**Figure 1 nutrients-11-01264-f001:**
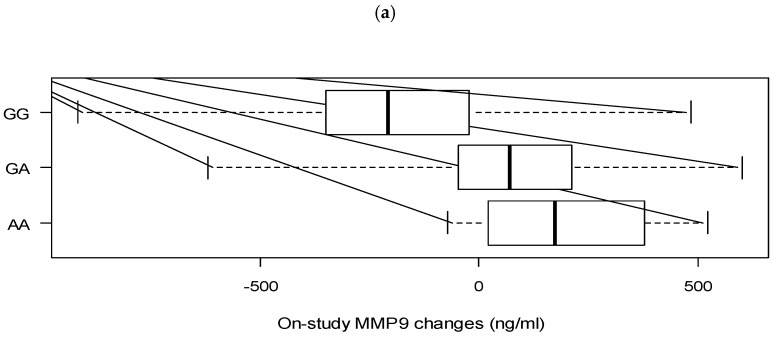
During intervention inflammatory biomarker changes in response to vitamin D3 supplementation (4000 IU/day for 12 weeks) across the *VDR* polymorphic groups. (**a**) Matrix metallopeptidase 9 (MMP9) changes during the study across the *VDR Cdx2* polymorphic groups; (**b**) MMP9 changes during the study across the *VDR TaqI* polymorphic groups; (**c**) Tumor necrosis factorα (TNFα) changes during the study across the *VDR BsmI* polymorphic groups.

**Table 1 nutrients-11-01264-t001:** Basic characteristics of the study participants.

Characteristic	Total
Age (years)	49.0 ± 8.8
Marital status, N (%)	
Married	137 (78)
Widow	10 (6)
Divorced	11 (6)
Single	18 (10)
Current smoking, N (%)	4 (2)
Hormone receptor status, N (%)	
ER+	102 (57.9)
ER+PR+	95 (53.9)
ER+PR-	6 (3.4)
ER-PR-	39 (22.1)
HER2+	17 (9.6)
ER-PR-HER2-	20 (11.3)
Family history of breast cancer, N (%)	30 (17)
Post-menopausal, N (%)	127 (72)
Hormone therapy for breast cancer	137 (78)
Regional nodal involvement, N (%)	
Non	0 (0)
1–3	33 (19)
4–9	44 (25)
≥10	73 (41)
Missing	26 (15)
Stage, N (%)	
Stage I	34 (23)
Stage II	75 (44)
Stage III	42 (27)
Missing	25 (6)
Recurrence, N (%)	
Yes	4 (2)
No	92 (52)
Unknown	80 (46)
Physical activity levels, N (%)	
Low physical activity	92 (52)
Moderate physical activity	49 (28)
High physical activity	34 (19)
Energy intake, Kcal	1906 (1835–2046)
BMI	28.0 (25.8–30.8)
*ApaI* N (%)	
AA	83 (47)
Aa	68 (39)
aa	25 (14)
*TaqI* N (%)	
TT	70 (40)
Tt	89 (51)
tt	17 (10)
*BsmI* N (%)	
BB	82 (47)
Bb	74 (42)
bb	20 (11)
*FokI* N (%)	
FF	82 (47)
Ff	72 (41)
ff	22 (12)
*Cdx2* N (%)	
GG	83 (47)
AG	68 (39)
AA	25 (14)

ER: Estrogen Receptor; PR: Progesterone Receptor; HER2: Human Epidermal Growth Factor Receptor 2; BMI: Body Mass Index. Values with normal distribution are presented as mean ± SD; values with non-normal distribution are presented as median (Q1, Q3) and categorical variables are presented as N (%).

**Table 2 nutrients-11-01264-t002:** Comparison of changes in response variables before and after vitamin D3 supplementation (4000 IU/day) for 12 weeks.

Variables	Before Intervention (*n* = 176)	After Intervention (*n* = 176)	On-Study Changes (*n* = 176)
25(OH)D (nmol/L)	41.5 ± 27.5	113.1 ± 45.8	71.5 (65.3, 77.7)
E-cadherin (ng/mL)	120.4 ± 63.4	112.8 ± 60.0	−7.66 (−10.62, −4.71)
MMP9 (ng/mL)	1746 ± 654.2	1655 ± 628.3	−90.9 (−134.4, −47.4)
IFNβ (pg/mL)	574.2 ± 241.5	578.8 ± 235.5	4.64 (−18.6, 27.9)
SICAM1 (ng/mL)	672.5 ± 228.1	650.8 ± 229.0	−21.7 (−36.0, −7.41)
SVCAM1 (ng/mL)	800.7 ± 235.6	797.3 ± 247.5	−3.47 (−17.3, 10.4)
IL6 (pg/mL)	6.10 ± 3.30	6.03 ± 3.23	−0.07 (−0.20, 0.04)
TNFα (pg/mL)	20.6 ± 8.90	19.3 ± 8.68	−1.32 (−1.78, −0.85)
PAI-1 (ng/mL)	46.2 ± 15.2	44.4 ± 15.5	−1.77 (−2.64, −0.91)
hs-CRP (mg/L)	2.69 ± 1.13	2.55 ± 1.04	−0.13 (−0.19, −0.06)

Before and after intervention values are presented as mean ± standard deviation (SD) and on-study changes as mean 95% confidence interval (CI). On-study variables changes were calculated by subtracting the value of pre- from the value of post intervention. 25(OH)D, 25-hydroxy vitamin D; MMP9, matrix metallopeptidase 9; IFNβ, interferon β; SICAM-1, soluble intercellular adhesion molecule-1; VCAM-1, soluble vascular cell adhesion molecule-1, IL6, interleukin 6 (IL6); TNFα, tumor necrosis factorα; PAI-1, plasminogen activator inhibitor-1, hs-CRP, human high sensitivity C-reactive protein.

**Table 3 nutrients-11-01264-t003:** Association analyses between vitamin D receptor single nucleotide polymorphisms (*VDR* SNP) genotypes and on-study breast cancer biomarkers changes among patients supplemented with vitamin D3 (4000 IU/day for 12 weeks) under codominant model.

Variables Changes	*ApaI* β (95% CI)			*TaqI* β (95% CI)			*BsmI* β (95% CI)			*FokI* β (95% CI)			*Cdx2* β (95% CI)		
	AA (*n* = 83)	Aa (*n* = 68)	aa (*n* = 25)	TT (*n* = 70)	Tt (*n* = 89)	tt (*n* = 17)	BB (*n* = 82)	Bb (*n* = 74)	bb (*n* = 20)	FF (*n* = 82)	Ff (*n* = 72)	ff (*n* = 22)	GG (*n* = 102)	GA (*n* = 60)	AA (*n* = 14)
Ecadherin (ng/mL) *p* value FDR *p* value *	Ref (0) 0.630.92	−0.06 (−0.22,0.09)	0.02 (−0.18,0.24)	Ref (0) 0.870.98	0.00 (−0.14, 0.16)	−0.05 (−0.31, 0.20)	Ref (0) 0.71 0.93	−0.06 (−0.21, 0.09)	−0.01 (−0.25, 0.22)	Ref (0) 0.94 0.98	0.02 (−0.13, 0.17)	0.02 (−0.20, 0.25)	Ref (0) 0.82 0.98	0.04 (−0.10, 0.20)	0.04 (−0.22, 0.31)
MMP9 (ng/mL) *p* value FDR *p* value *	Ref (0) 0.26 0.87	0.15 (−0.08, 0.39)	−0.08 (−0.40, 0.24)	Ref (0) 0.008 0.15	−0.01 (−0.24, 0.20)	−0.58 (−0.95, −0.20)	Ref (0) 0.08 0.57	−0.23 (−0.45, −0.00)	−0.29 (−0.65, 0.05)	Ref (0) 0.13 0.73	−0.19 (−0.42, 0.03)	−0.27 (−0.61, 0.07)	Ref (0) <0.001 0.001	0.45 (0.23, 0.68)	0.60 (0.21, 0.99)
IFNβ (pg/mL) *p* value FDR *p* value *	Ref (0) 0.96 0.98	−0.03 (−0.27, 0.20)	−0.01 (−0.34, 0.31)	Ref (0) 0.53 0.92	0.13 (−0.09, 0.36)	0.06 (−0.32, 0.45)	Ref (0) 0.48 0.92	0.05 (−0.17, 0.28)	0.22 (−0.14, 0.58)	Ref (0) 0.06 0.54	−0.05 (−0.28, 0.17)	−0.40 (−0.75, −0.06)	Ref (0) 0.66 0.92	0.07 (−0.15, 0.31)	0.15 (−0.25, 0.57)
s−ICAM-1 (ng/mL) *p* value FDR *p* value *	Ref (0) 0.84 0.98	6.55 (−25.3, 38.4)	11.7 (−31.7, 55.2)	Ref (0) 0.39 0.87	20.0 (−10.3, 50.4)	21.9 (−29.3, 73.2)	Ref (0) 0.28 0.87	−8.6 (−39.1, 21.9)	−38.9 (−86.6, 8.8)	Ref (0) 0.95 0.98	−1.4 (−32.3, 29.5)	−6.9 (−53.0, 39.1)	Ref (0) 0.02 0.22	29.1 (−1.7, 59.9)	63.8 (10.4, 117.2)
s-VCAM-1 (ng/mL) *p* value FDR *p* value *	Ref (0) 0.29 0.87	0.15 (−0.08, 0.40)	0.21 (−0.11, 0.54)	Ref (0) 0.09 0.57	−0.19 (−0.42, 0.03)	−0.37 (−0.76, 0.01)	Ref (0) 0.45 0.92	−0.00 (−0.23, 0.23)	−0.22 (−0.58, 0.14)	Ref (0) 0.19 0.87	−0.11 (−0.35, 0.11)	−0.31 (−0.66, 0.03)	Ref (0) 0.86 0.98	0.03 (−0.20, 0.27)	0.10 (−0.31, 0.52)
IL6 (pg/mL) *p* value FDR *p* value *	Ref (0 0.20 0.87	−0.08 (−0.35, 0.18)	0.25 (−0.10, 0.62)	Ref (0) 0.37 0.87	−0.18 (−0.44, 0.07)	−0.08 (−0.51, 0.35)	Ref (0) 0.22 0.87	−0.10 (−0.35, 0.15)	−0.35 (−0.76, 0.04)	Ref (0) 0.32 0.87	−0.07 (−0.33, 0.18)	0.22 (−0.16, 0.61)	Ref (0) 0.69 0.93	−0.05 (−0.32, 0.21)	0.15 (−0.31, 0.61)
TNFα (pg/mL) *p* value FDR *p* value *	Ref (0) 0.77 0.98	−0.01 (−0.13, 0.09)	−0.05 (−0.20, 0.09)	Ref (0) 0.58 0.92	0.05 (−0.05, 0.15)	0.06 (−0.11, 0.24)	Ref (0) 0.01 0.15	−0.01 (−0.11, 0.09)	−0.24 (−0.40, −0.08)	Ref (0) 0.92 0.98	0.02 (−0.08, 0.13)	0.01 (−0.14, 0.17)	Ref (0) 0.55 0.92	−0.05 (−0.16, 0.05)	−0.04 (−0.23, 0.14)
PAI-1 (ng/mL) *p* value FDR *p* value *	Ref (0 0.62 0.92	0.04 (−0.07, 0.15)	0.06 (−0.08, 0.22)	Ref (0) 0.99 0.99	0.00 (−0.10, 0.11)	−0.00 (−0.18, 0.18)	Ref (0) 0.54 0.92	0.04 (−0.06, 0.15)	−0.03 (−0.20, 0.13)	Ref (0) 0.65 0.92	−0.00 (−0.11, 0.10)	−0.07 (−0.24, 0.08)	Ref (0) 0.33 0.87	0.03 (−0.07, 0.15)	0.14 (−0.05, 0.33)
hs-CRP (mg/L) *p* value FDR *p* value *	Ref (0)0.42 0.90	0.03 (−0.02, 0.08)	0.04 (−0.03, 0.12)	Ref (0 0.36 0.87	−0.03 (−0.09, 0.01)	−0.00 (−0.09, 0.08)	Ref (0) 0.90 0.98	−0.00 (−0.05, 0.05)	−0.01 (−0.10, 0.06)	Ref (0) 0.27 0.87	0.04 (−0.01, 0.09)	0.04 (−0.03, 0.12)	Ref (0) 0.65 0.92	−0.02 (−0.08, 0.03)	0.00 (−0.09, 0.10)

SNP Single Nucleotide Polymorphism; MMP9, matrix metallopeptidase 9; IFNβ, interferon β; SICAM-1, soluble intercellular adhesion molecule-1; VCAM-1, soluble vascular cell adhesion molecule-1, IL6, interleukin 6 (IL6); TNFα, tumor necrosis factorα; PAI-1, plasminogen activator inhibitor-1, hs-CRP, human high sensitivity C-reactive protein. β is standardized regression coefficient with 95% confidence interval. On-study variables changes were calculated by subtracting the value of pre- from the value of post intervention. Models were adjusted for age, 25-hydroxy vitamin D (25(OH)D) and body mass index (BMI) at baseline. * FDR *p* value resulted from the false-discovery rate (FDR) methods.

**Table 4 nutrients-11-01264-t004:** Association analyses between *VDR* SNP genotypes and on-study breast cancer biomarkers changes among patients supplemented with vitamin D3 (4000 IU/day for 12 weeks) under recessive model.

Variables Changes	*ApaI* β (95% CI)		*TaqI* β (95% CI)		*BsmI* β (95% CI)		*FokI* β (95% CI)		*Cdx2* β (95% CI)	
	AA & Aa (*n* = 151)	aa (*n* = 25)	TT & Tt (*n* = 159)	tt (*n* = 17)	BB & Bb (*n* = 156)	bb (*n* = 20)	FF & Ff (*n* = 154)	ff (*n* = 22)	GG & GA (*n* = 162)	AA (*n* = 14)
E-cadherin (ng/mL) *p* value FDR *p* value *	Ref (0) 0.58 0.91	0.05 (−0.14, 0.26)	Ref (0) 0.61 0.91	−0.06 (−0.30, 0.18)	Ref (0) 0.86 0.99	0.01 (−0.20, 0.24)	Ref (0) 0.98 0.99	0.01 (−0.19, 0.23)	Ref (0) 0.85 0.99	0.02 (−0.23, 0.28)
MMP9 (ng/mL) *p* value FDR *p* value *	Ref (0) 0.32 0.84	−0.15 (−0.46, 0.15)	Ref (0) 0.001 0.04	−0.57 (−0.92, −0.21)	Ref (0) 0.29 0.81	−0.18 (−0.52, 0.15)	Ref (0) 0.28 0.81	−0.17 (−0.50, 0.14)	Ref (0) 0.03 0.38	0.42 (0.03, 0.82)
IFNβ (pg/mL) *p* value FDR *p* value *	Ref (0) 0.99 0.99	−0.00 (−0.31, 0.31)	Ref (0) 0.94 0.99	−0.01 (−0.38, 0.35)	Ref (0) 0.26 0.81	0.19 (−0.14, 0.53)	Ref (0) 0.02 0.30	−0.38 (−0.70,−0.05)	Ref (0) 0.53 0.91	0.12 (−0.27, 0.52)
s-ICAM-1 (ng/mL) *p* value FDR *p* value *	Ref (0) 0.67 0.91	8.89 (−32.1, 49.9)	Ref (0) 0.66 0.91	10.7 (−37.8, 59.2)	Ref (0) 0.13 0.65	−34.6 (−79.8, 10.5)	Ref (0) 0.77 0.96	−6.2 (−49.9, 37.3)	Ref (0) 0.05 0.45	52.7 (0.27, 105.1)
s-VCAM-1 (ng/mL) *p* value FDR *p* value *	Ref (0) 0.36 0.85	0.14 (−0.16, 0.45)	Ref (0) 0.15 0.65	−0.26 (−0.63, 0.10)	Ref (0) 0.21 0.72	−0.22 (−0.56, 0.12)	Ref (0) 0.12 0.65	−0.25 (−0.59, 0.07)	Ref (0) 0.66 0.91	0.09 (−0.31, 0.49)
IL6 (pg/mL) *p* value FDR *p* value *	Ref (0) 0.09 0.65	0.29 (−0.05, 0.64)	Ref (0) 0.91 0.99	0.02 (−0.39, 0.43)	Ref (0) 0.11 0.65	−0.30 (−0.69, 0.07)	Ref (0) 0.16 0.65	0.26 (−0.10, 0.63)	Ref (0) 0.45 0.91	0.17 (−0.27, 0.62)
TNFα (pg/mL) *p* value FDR *p* value *	Ref (0) 0.52 0.91	−0.04 (−0.18, 0.09)	Ref (0) 0.72 0.95	0.03 (−0.13, 0.20)	Ref (0) 0.002 0.04	−0.24 (−0.39,−0.08)	Ref (0) 0.98 0.99	0.00 (−0.15, 0.15)	Ref (0) 0.81 0.98	−0.02 (−0.20, 0.16)
PAI-1 (ng/mL) *p* value FDR *p* value *	Ref (0) 0.49 0.91	0.05 (−0.09, 0.19)	Ref (0) 0.94 0.99	−0.00 (−0.18, 0.16)	Ref (0) 0.46 0.91	−0.06 (−0.22, 0.10)	Ref (0) 0.36 0.85	−0.07 (−0.22, 0.08)	Ref (0) 0.18 0.67	0.12 (−0.06, 0.31)
Hs-CRP (mg/L) *p* value FDR *p* value *	Ref (0) 0.42 0.91	0.02 (−0.04, 0.10)	Ref (0) 0.67 0.91	0.01 (−0.06, 0.10)	Ref (0) 0.65 0.91	−0.01 (−0.09, 0.06)	Ref (0) 0.50 0.91	0.02 (−0.05, 0.10)	Ref (0) 0.74 0.95	0.01 (−0.07, 0.10)

SNP Single Nucleotide Polymorphism; MMP9, matrix metallopeptidase 9; IFNβ, interferon β; SICAM-1, soluble intercellular adhesion molecule-1; VCAM-1, soluble vascular cell adhesion molecule-1, IL6, interleukin 6 (IL6); TNFα, tumor necrosis factorα; PAI-1, plasminogen activator inhibitor-1, hs-CRP, human high sensitivity C-reactive protein. β is standardized regression coefficient with 95% confidence interval. On-study variables changes were calculated by subtracting the value of pre- from the value of post intervention. Models were adjusted for age, 25-hydroxy vitamin D (25(OH)D) and body mass index (BMI) at baseline. * FDR *p* value resulted from the false-discovery rate (FDR) methods.

**Table 5 nutrients-11-01264-t005:** Haplotype block analysis of variable changes after vitamin D supplementation (4000 IU/day) for 12 weeks.

Variables Changes	H1			H2			H3			H4			H5		
	Global-Stat	*p* Value *	FDR *p* Value **	Global-Stat	*p* Value *	FDR *p* Value **	Global-Stat	*p* Value *	FDR *p* Value **	Global-Stat	*p* Value *	FDR *p* Value **	Global-Stat	*p* Value *	FDR *p* Value **
E-cadherin	11.46	0.90	0.97	1.78	0.97	0.99	0.81	0.99	0.99	4.38	0.73	0.96	2.29	0.89	0.97
MMP9	56.20	<0.001	<0.001	21.20	0.003	0.04	4.81	0.68	0.95	12.58	0.08	0.45	50.40	<0.001	<0.001
IFNβ	13.45	0.89	0.97	7.84	0.35	0.83	3.99	0.77	0.97	5.16	0.63	0.91	3.10	0.79	0.97
s-ICAM-1	31.15	0.03	0.22	16.47	0.02	0.22	6.86	0.44	0.83	8.01	0.33	0.83	13.45	0.03	0.22
s-VCAM-1	11.37	0.91	0.97	7.59	0.36	0.83	4.39	0.73	0.96	5.57	0.59	0.91	5.61	0.46	0.83
IL6	15.07	0.81	0.97	9.53	0.21	0.78	6.97	0.43	0.83	6.33	0.50	0.83	7.56	0.27	0.83
TNFα	21.96	0.28	0.78	10.43	0.16	0.78	7.76	0.35	0.83	8.89	0.26	0.83	11.32	0.07	0.45
PAI-1	20.27	0.50	0.83	5.38	0.61	0.91	0.59	0.99	0.99	2.78	0.90	0.97	5.44	0.48	0.83
hs-CRP	24.08	0.19	0.78	7.11	0.41	0.83	5.29	0.62	0.91	6.96	0.43	0.83	5.52	0.47	0.83

Haplotype 1(H1), *Cdx2*, *FokI*, *BsmI*, *ApaI*, *TaqI*; Haplotype 2(H2), *Cdx2*, *FokI*, *BsmI*, Haplotype 3(H3), *BsmI*, *ApaI*, *TaqI*; Haplotype 4(H4), *FokI*, *BsmI*, *ApaI*; Haplotype 5(H5), *FokI*, *TaqI*, *Cdx2*. MMP9, matrix metallopeptidase 9; IFNβ, interferon β; s-ICAM-1, soluble intercellular adhesion molecule-1; s-VCAM-1, soluble vascular cell adhesion molecule-1, IL6, interleukin 6 (IL6); TNFα, tumor necrosis factorα; PAI-1, plasminogen activator inhibitor-1, hs-CRP, human high sensitivity C-reactive protein. On-study variables changes were calculated by subtracting the value of pre- from the value of post intervention. * Adjusted for age, baseline 25-hydroxy vitamin D (25(OH)D) and body mass index (BMI). ** *p* value resulted from the false-discovery rate (FDR) methods.

**Table 6 nutrients-11-01264-t006:** Causal mediation analysis of the association between *VDR* SNPs and inflammatory biomarkers by inclusion of on-study plasma 25-hydroxy vitamin D (25(OH)D changes as mediator.

Outcomes	Exposures	ACME * (95% CI)		ADE ** (95% CI)		Total Effect (95% CI)		Proportion Mediated 95% CI	
**MMP9** (ng/mL)	*TaqI*	tt	Tt	tt	Tt	tt	Tt	tt	Tt
		-34.904 (−84.45, −2.74)	−9.22 (−31.47, 8.80)	−12.64 (−164.58, 135.89)	−8.70 (−95.09, 84.81)	−47.54 (−201.15, 110.50)	−17.92 (−106.64, 76.60)	0.29 (−3.31, 6.10)	0.11 (−2.45, 2.89)
	*Cdx2*	GG	GA	GG	GA	GG	GA	GG	GA
		14.85 (−10.99, 54.72)	5.34 (−10.46, 23.17)	400.51 (275.52, 529.81)	276.29 (196.69, 355.65)	415.37 (283.69, 551.44)	281.64 (199.56, 361.93)	0.03 (−0.03, 0.12)	0.01 (−0.04, 0.08)
**TNFα** (pg/mL)	*BsmI*	bb	Bb	bb	Bb	bb	Bb	bb	Bb
		−0.003 (−0.13, 0.12)	−0.21 (−0.66, 0.12)	−0.11 (−1.03, 0.82)	−1.53 (−3.04, 0.03)	−0.11 (−1.04, 0.83)	−1.75 (−3.22, −0.22)	0.007 (−1.20, 1.35)	0.11 (−0.10, 0.82)

* Average causal mediation effects, ** average direct effect. Adjusted for age, baseline 25-hydroxy vitamin D (25(OH)D) and body mass index (BMI). MMP9, matrix metallopeptidase 9; TNFα, tumor necrosis factorα. TT, GG and BB were considered as reference category for *TaqI*, *Cdx2* and *BsmI,* respectively.

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
