# Peer review of "Vitamin D Receptor Genetic Variation and Cancer Biomarkers among Breast Cancer Patients Supplemented with Vitamin D3: A Single-Arm Non-Randomized Before and After Trial"

_nutrients, 2019, doi:10.3390/nu11061264_

Round 1

Reviewer 1 Report

Authors have greatly improved the manucript. I have only minor concerns on the work:

line 120: The authors estate that " detailed study protocol was described elsewhere." A reference is required.

line 121: do not delete "were admitted"

lines 206-209: How were these determined? Also by ELISA?

Conclusions: I would have preferred  more direct statements by changing indefinite words like "certain"  (line 459) or "other" (line 462) by the name of the biomarkers.

Author Response

We thank the reviewers for their careful reading of the manuscript and especially for their helpful comments. In this revision every attempt has been made to retain these positive features and improve the presentation in terms of both clarity and scientific rigor. We will now address the specific comments of each reviewer.

Reviewer#1

Authors have greatly improved the manuscript. I have only minor concerns on the work:

Comment: line 120: The authors estate that " detailed study protocol was described elsewhere." A reference is required.

Response: It has been added.

Comment: line 121: do not delete "were admitted"

Response: It has been corrected to “who were admitted to” (Page 3, line 109)

Comment: lines 206-209: How were these determined? Also by ELISA?

Response: Yes, ELISA was used to determine plasma inflammatory biomarkers. It has been added to the text. (Page 5, line 186)

Comment: Conclusions: I would have preferred more direct statements by changing indefinite words like "certain"  (line 459) or "other" (line 462) by the name of the biomarkers.

Response: Agreed and revised. (Page 10 line 414-415; Page 10, line 416)

Reviewer 2 Report

The authors have incorporated all my comments and those of reviewer 2.

Author Response

Reviewer#2

 Comment: The authors have incorporated all my comments and those of reviewer 2. 

Response: We appreciate your precise review to improve our manuscript

This manuscript is a resubmission of an earlier submission. The following is a list of the peer review reports and author responses from that submission.

Round 1

Reviewer 1 Report

This is a very interesting study and comprehensive analysis of the association of VDR genetic variants with the prognosis (represented as the change in inflammatory and tumor markers) among breast cancer patients that had been supplemented with vitamin D. I have major and minor comments that need addressing before the paper can be considered for publication.

Major comments

1)      I think the general pattern in the analyses of the association of VDR SNPs and tumor markers (Table 3) is of no association, except perhaps for some associations of TaqI and CdX2 with MMP9, some of which were not significant after correcting for multiple testing. The associations with haplotypes were inconsistent and only limited to some allele combinations. In general too much emphasis is placed on the significant results observed in this possibly underpowered study, while obviating the lack of association for most of the SNPs and tumor markers.

Therefore, I believe the title, the conclusion sentence in the abstract (lines 44-46), the discussion (first paragraph), and conclusion, are misleading and should be reworded. For the title I suggest: “Vitamin D Receptor Genetic Variation and Cancer Biomarkers among Breast Cancer Patients supplemented with Vitamin D3: a single-arm non-randomized before and after trial” For the abstract conclusion I suggest: “Among breast cancer patients who received vitamin D3 supplementation, VDR genetic polymorphisms were not associated with longitudinal changes in cancer biomarkers”.

2)      The study design is a single-arm non-randomized before and after trial, but this is not specified in the title nor abstract. However, the analyses are purely observational, as no randomisation of the exposure (genetic variability) is possible. What the authors did is just to observe and describe the relationship between genetic variability and tumor markers among a highly selected group of patients who had received vitamin D supplementation. The lack of a comparator group makes it very difficult to attribute any longitudinal changes in tumour markers to vitamin D only, as these markers could have also improved over time without vitamin D.

I would rather describe these analyses as an observational prospective cohort study of patients receiving vitamin D3 supplementation. Or as an alternative, an observational secondary analysis of a single-arm non-randomized before and after trial.

3)      I found it surprising that survival information was available, but only mentioned in one sentence in the results (line 221). Have the authors published these data already and have they related them to genetic polymorphisms?

4)      While I was able to find some typos, wording and grammar errors (see minor comments below), I think that overall the manuscript needs a native English speaker to proofread it.

Minor comments

1)      Introduction, line 56: “which (are) related to”

2)      Introduction, lines 74 to 76: “no controlled trial has been performed to investigate the possible biological interactive effects of the VDR genetic variations and vitamin D intake on diverse aspects of breast cancer host response.”

3)      Introduction, lines 86 to 88: “Future trials using genetic 86 and genomic testing along with vitamin D levels will inform, classify risk and results personalized 87 follow up to primary cancer prevention and treatment.”

4)      Introduction, lines 88 to 90: “To address this gap in our literature, for the 88 first time, we investigated whether VDR polymorphisms (ApaI, TaqI, FokI, BsmI and Cdx2) could affect…” Is there reliable evidence to support that these SNPs determine survival of breast cancer patients?

5)      Study design: please break this section into separate paragraphs and consider using additional headings to distinguish between the intervention provided (lines 111 to 114, and lines 121 to 126), outcomes (lines 115-121), and covariate assessment (lines 127 to 129). For the latter (covariate assessment), consider combining with study measurements.

6)      Methods, line 125: “Adherence was calculated…”

7)      Methods, line 151: “Subjects were requested…”

8)      Methods, line 175: Is Austeria the name of the company or do you mean Austria/Australia?

9)      Line 181: “to test the (normality of the) distribution of (continuous) variables”

10)  Line 182: I believe the test used was Mann-Whitney U test or Wilcoxon rank sum test for unpaired samples, to distinguish from Wilcoxon signed-rank test (for paired samples).

11)  Lines 213 to 215: which analyses were done in Stata and which in R?

12)  Table 1: Please left align Characteristics column and add indentation for values within categorical variables.

13)  Line 235: 25(OH)D units reported as “nmol/liter”. Also in Table 2, units reported as “nmol/ml”. 25(OH)D units should be “mmol/L” or “ng/ml”.

14)  Table 2: please report just 3 significant figures (i.e. two decimals for numbers less than 10, and just one decimal for numbers above or equal to 10). Change in Ecadherin is not plausible (99-98 cannot be equal to -7.49). Same for most of the variables. Although I am aware that the data are not normally distributed, and the authors are reporting the median and interquartile range, in the differences and p-values when looking at the values before and after the intervention. The sample size is larger than 100, so I would suggest using the means and SDs anyways to show the data before and after the intervention, and calculate mean differences and 95% CIs for the on-study change.

15)  Line 278: wording is incorrect. Please re-write as “…haplotypes which did not remain statistically…” or “…haplotypes which remained statistically…” as applicable.

16)  In Table 3 for some of the tumor markers in the reference category for some SNPs it shows “Ref (0)” while in other cases it shows only “0”

17)  Line 369: “…among participants (of) in the current study…”

18)  Line 371: “One limitation of our study was the small number…”

19)  Figure S1, please fix the arrows in the flowchart (at the moment they appear to be crooked).

Author Response

We thank the reviewers for their careful reading of the manuscript and especially for their helpful comments. In this revision every attempt has been made to retain these positive features and improve the presentation in terms of both clarity and scientific rigor. We will now address the specific comments of each reviewer. Please also see the attached file.

Reviewer#1

This is a very interesting study and comprehensive analysis of the association of VDR genetic variants with the prognosis (represented as the change in inflammatory and tumor markers) among breast cancer patients that had been supplemented with vitamin D. I have major and minor comments that need addressing before the paper can be considered for publication.

Major comments

1)          I think the general pattern in the analyses of the association of VDR SNPs and tumor markers (Table 3) is of no association, except perhaps for some associations of TaqI and CdX2 with MMP9, some of which were not significant after correcting for multiple testing. The associations with haplotypes were inconsistent and only limited to some allele combinations. In general too much emphasis is placed on the significant results observed in this possibly underpowered study, while obviating the lack of association for most of the SNPs and tumor markers. Therefore, I believe the title, the conclusion sentence in the abstract (lines 44-46), the discussion (first paragraph), and conclusion, are misleading and should be reworded. For the title I suggest: “Vitamin D Receptor Genetic Variation and Cancer Biomarkers among Breast Cancer Patients supplemented with Vitamin D3: a single-arm non-randomized before and after trial” For the abstract conclusion I suggest: “Among breast cancer patients who received vitamin D3 supplementation, VDR genetic polymorphisms were not associated with longitudinal changes in cancer biomarkers”.

Response: Thank you for the suggestion. Agreed and revised in the title, abstract (line 51-55), discussion (page 8, line 321-326) and conclusions (page 10, line 426-429).

2)          The study design is a single-arm non-randomized before and after trial, but this is not specified in the title nor abstract. However, the analyses are purely observational, as no randomisation of the exposure (genetic variability) is possible. What the authors did is just to observe and describe the relationship between genetic variability and tumor markers among a highly selected group of patients who had received vitamin D supplementation. The lack of a comparator group makes it very difficult to attribute any longitudinal changes in tumour markers to vitamin D only, as these markers could have also improved over time without vitamin D. I would rather describe these analyses as an observational prospective cohort study of patients receiving vitamin D3 supplementation. Or as an alternative, an observational secondary analysis of a single-arm non-randomized before and after trial.

Response: Thank you for your thoughtful comment. This issue is addressed in the revised version of manuscript in the title, abstract (line 40) and methods (Page 3 line 124).

3)          I found it surprising that survival information was available, but only mentioned in one sentence in the results (line 221). Have the authors published these data already and have they related them to genetic polymorphisms?

Response: Sorry for not making it clear. Survival was not assessed in this study. In the current investigation, the outcomes of interest were inflammatory and immune biomarkers as well as those associated with cell proliferation, differentiation and metastasis. We aimed to looking at the potentials mechanisms (by measuring cancer biomarkers) through which vitamin D involved in carcinogenesis. As mentioned in the methods section, this study was conducted in breast cancer survivors who were admitted to Shohaday-e-Tajrish hospital and its associated clinics for the cancer follow up. Patients were diagnosed with breast cancer at least 6 months before the study enrollment and treatment protocol including surgery, radio and chemotherapy was completed requested to participate in the study. The average survival time for the study subjects from diagnosis to recruitment was 4.5 years (range 1– 16). It is clarified in the results (page 6 line 248).

4)          While I was able to find some typos, wording and grammar errors (see minor comments below), I think that overall the manuscript needs a native English speaker to proofread it.

-Response: The manuscript checked and edited by a native English speaker.

Minor comments

1)          Introduction, line 56: “which (are) related to”

-Response: it was edited in the text. (Page 2, line 65)

2)      Introduction, lines 74 to 76: “no controlled trial has been performed to investigate the possible biological interactive effects of the VDR genetic variations and vitamin D intake on diverse aspects of breast cancer host response.”

-Response: it was edited in the text. (Page 2, line 85-87)

3)      Introduction, lines 86 to 88: “Future trials using genetic 86 and genomic testing along with vitamin D levels will inform, classify risk and results personalized 87 follow up to primary cancer prevention and treatment.”

-Response: it was amended in the text. (Page 3, line 98-100)

4)      Introduction, lines 88 to 90: “To address this gap in our literature, for the 88 first time, we investigated whether VDR polymorphisms (ApaI, TaqI, FokI, BsmI and Cdx2) could affect…” Is there reliable evidence to support that these SNPs determine survival of breast cancer patients?

-Response: Yes, results of previous studies have suggested that variation in VDR was associated with cancer mortality and survival [1]. This point is added in the introduction (page 2, line 82-83) in the revised manuscript.

1. Mondul, A.M., et al., Vitamin D and Cancer Risk and Mortality: State of the Science, Gaps, and Challenges. Epidemiol Rev, 2017. 39(1): p. 28-48.

5)      Study design: please break this section into separate paragraphs and consider using additional headings to distinguish between the intervention provided (lines 111 to 114, and lines 121 to 126), outcomes (lines 115-121), and covariate assessment (lines 127 to 129). For the latter (covariate assessment), consider combining with study measurements.

-Response: Agreed and done. Page 3 line 122; page 3 line 134; page 4 line 155

6)      Methods, line 125: “Adherence was calculated…”

-Response: it was corrected. (Page 3 line 131)

7)      Methods, line 151: “Subjects were requested…”

-Response: it was amended. (Page 4 line 176)

8)      Methods, line 175: Is Austeria the name of the company or do you mean Austria/Australia?

-Response: Sorry for the misspelling. it was Austria and edited in the text. page 5 line 200-201

9)      Line 181: “to test the (normality of the) distribution of (continuous) variables”

-Response: it was corrected. (Page 5, line 206-207)

10)  Line 182: I believe the test used was Mann-Whitney U test or Wilcoxon rank sum test for unpaired samples, to distinguish from Wilcoxon signed-rank test (for paired samples).

-Response: Sorry for not clarifying this point in the text. It is clarified in the revised version. (Page 5, line 208)

11)  Lines 213 to 215: which analyses were done in Stata and which in R?

Response: W used Stata to compare the difference between groups and also descriptive statistics, while the other analyses was carried out in R.

12)  Table 1: Please left align Characteristics column and add indentation for values within categorical variables.

-Response: Done. (Table 1)

13)  Line 235: 25(OH)D units reported as “nmol/liter”. Also in Table 2, units reported as “nmol/ml”. 25(OH)D units should be “mmol/L” or “ng/ml”.

-Response: Sorry for the typing error. It was corrected.

14)  A) Table 2: please report just 3 significant figures (i.e. two decimals for numbers less than 10, and just one decimal for numbers above or equal to 10).

Response: Done

B) Change in Ecadherin is not plausible (99-98 cannot be equal to -7.49). Same for most of the variables. Although I am aware that the data are not normally distributed, and the authors are reporting the median and interquartile range, in the differences and p-values when looking at the values before and after the intervention. The sample size is larger than 100, so I would suggest using the means and SDs anyways to show the data before and after the intervention, and calculate mean differences and 95% CIs for the on-study change.

Response: Due to highly skewed distribution of our variables, we have to report them as median (IQR). We rechecked the E-cadherin and the results remain unchanged because the median of difference is not equal to the difference of the medians.

The mean and SD value of variables reported here for the reviewer. Also, the histogram for some variable are provided here. This shape of histogram occurs for almost all of variables. However, if the reviewer still suggest the mean and SD is a better representation of variables, this table could be transferred to the manuscript.   

Table 2. Comparison of changes in response variables before and after vitamin D3 supplementation (4000 IU/day) for 12 weeks.

Variables

Before   intervention

(n=176)

After   intervention

(n=176)

On-study   changes

(n=176)

P   value*

25(OH)D (mmol/ml)

41.5   ± 27.5

113.1   ± 45.8

71.5 (65.3,77.7)

<0.001

E-cadherin(ng/ml)

120.4   ± 63.4

112.8   ± 60.0

-7.66 (-10.62, -4.71)

<0.001

MMP9(ng/mL)

1746   ± 654.2

1655   ± 628.3

-90.9 (-134.4, -47.4)

<0.001

IFNβ(pg/mL)

574.2   ± 241.5

578.8   ± 235.5

4.64 (-18.6, 27.9)

0.42

SICAM1(ng/ml)

672.5   ± 228.1

650.8   ± 229.0

-21.7 (36.0, -7.41)

0.006

SVCAM1(ng/ml)

800.7   ± 235.6

797.3 ± 247.5

-3.47 (-17.3, 10.4)

0.40

IL6 (pg/mL)

6.10   ± 3.30

6.03   ± 3.23

-0.07 (-0.20, 0.04)

0.31

TNFα(pg/mL)

20.6   ± 8.90

19.3   ± 8.68

-1.32 (-1.78, -0.85)

<0.001

PAI-1(ng/mL)

46.2   ± 15.2

44.4   ± 15.5

-1.77 (-2.64, -0.91)

<0.001

hs-CRP(mg/L)

2.69   ± 1.13

2.55   ± 1.04

-0.13 (-0.19, -0.06)

0.0003

*P values were calculated using Wilcoxon signed-rank test.

 Before and after intervention values are presented as mean ± standard deviation (SD) and on study changes as mean 95% confidence interval (CI).

25(OH)D, 25-hydroxy vitamin D; MMP9, matrix metallopeptidase 9; IFNβ, interferon β; SICAM-1, soluble intercellular adhesion molecule-1; VCAM-1, soluble vascular cell adhesion molecule-1, IL6, interleukin 6 (IL6); TNFα, tumor necrosis factorα; PAI-1, plasminogen activator inhibitor-1, hs-CRP, human high sensitivity C-reactive protein.

15)  Line 278: wording is incorrect. Please re-write as “…haplotypes which did not remain statistically…” or “…haplotypes which remained statistically…” as applicable.

Response: Sorry for the mistake. It was corrected in the text. (Page 8 line 306)

16)  In Table 3 for some of the tumor markers in the reference category for some SNPs it shows “Ref (0)” while in other cases it shows only “0”

Response: It was corrected in the Table 3.

17)  Line 369: “…among participants (of) in the current study…”

Thank you for the careful reading of the manuscript. It was edited in the text. (Page 5, line 406).

18)  Line 371: “One limitation of our study was the small number…”

Response: it was edited. (Page 5, line 408).

19)  Figure S1, please fix the arrows in the flowchart (at the moment they appear to be crooked).

Response: Done.

Reviewer 2 Report

This is as a complete and well designed study, but I think that the changes in marker expression are so small as to doubt of their biological significance. Furthermore, the way these results are exposed is not clear cut, too standard and the results are not discussed in depth. For instance, Table 2 shows the changes in marker expression before and after supplementing with VitD. While the levels of 25(OH)D change as expected after treatment, the values for the markers are very small and similar in the two groups (as mean, Q1 or Q3) and this makes me suspect that the statistical difference after the WR test could be due to outliers. I would have thanked a graphical representation of the data or at least some discussion on this topic..   

There's no justification or discussion on why those markers were selected, neither on why the conspicuous inflammatory marker NF-kb was not chosen.

The discussion is also poor. It is stated several times that 25(OH)D treatment "suppresses the tumor suppressor E-Cadherin", but suppressing a tumor suppressor enhances tumorigenesis. I understand that authors wanted to state that 25(OH)D reduced levels of plasma E-Cad, likely by reinforcing adhesion of breast epithelial cells.

Several abbreviations are not defined (ACME, ADE), and the same for the B/F polymorphisms.   

Author Response

We thank the reviewers for their careful reading of the manuscript and especially for their helpful comments. In this revision every attempt has been made to retain these positive features and improve the presentation in terms of both clarity and scientific rigor. We will now address the specific comments of each reviewer. Please also see the attached file.

Reviewer # 2

This is as a complete and well designed study, but I think that

1)

A)    the changes in marker expression are so small as to doubt of their biological significance.

-Response: Thank you for your thoughtful comment. Based on your comment and also the comment of the reviewer 1 this issue is addressed in the title, abstract (line 53-57), discussion (page 8, line 326-331) and conclusions (page 10, line 434-437). of revised version of manuscript. Overall, we conclude that there is no association between cancer biomarkers and VDR polymorphisms except a possible association of certain inflammatory biomarkers and some VDR SNPs and haplotypes.

B)    Furthermore, the way these results are exposed is not clear cut, too standard and the results are not discussed in depth. For instance, Table 2 shows the changes in marker expression before and after supplementing with VitD. While the levels of 25(OH)D change as expected after treatment, the values for the markers are very small and similar in the two groups (as mean, Q1 or Q3) and this makes me suspect that the statistical difference after the WR test could be due to outliers. I would have thanked a graphical representation of the data or at least some discussion on this topic. 

- Response: Thank you for your thoughtful comment. The requested graphical representation of the data was provided here.

2) There's no justification or discussion on why those markers were selected, neither on why the conspicuous inflammatory marker NF-kb was not chosen.

-Response: Preclinical studies have suggested that 1,25(OH)2D mediates anti-carcinogenesis effects through suppression of inflammation, extracellular proteases, the tumor suppressor gene E-cadherin, adhesion molecules and down regulation of the NF-κβ signaling [28]; So, biomarkers of proposed pathways including E-cadherin, matrix metallopeptidase 9 (MMP9), interferon β (IFNβ), soluble intercellular adhesion molecule-1 (s-ICAM-1), soluble vascular cell adhesion molecule-1 (s-VCAM-1), tumor necrosis factorα (TNFα), interleukin 6 (IL6), plasminogen activator inhibitor-1(PAI-1), human high sensitivity C-reactive protein (hs-CRP) were chosen as outcomes of interest. (Page 3, line 139-145; page 9, line 343-344)

NF-kB acts as a central mediator of immune and inflammatory responses including NF-κB-dependent transcription of cytokines, chemokines, cell adhesion molecules and acute phase proteins. NF-kB has also been suggested as regulators of apoptosis and proliferation indicating its role in cell growth, proliferation, and survival. So the study markers were selected based on these pathways too[2].

2.         Oeckinghaus, A. and S. Ghosh, The NF-kappaB family of transcription factors and its regulation. Cold Spring Harbor perspectives in biology, 2009. 1(4): p. a000034-a000034

1)           

A)    The discussion is also poor.

Responses: Your comment about interpretation of results is right. Results of existing studies are not directly comparable with our work which might be the cause of this problem. However, we have tried to improve the interpretation of results in the revised manuscript. Any additional comments to improve the discussion section are most welcome.

B)    It is stated several times that 25(OH)D treatment "suppresses the tumor suppressor E-Cadherin", but suppressing a tumor suppressor enhances tumorigenesis. I understand that authors wanted to state that 25(OH)D reduced levels of plasma E-Cad, likely by reinforcing adhesion of breast epithelial cells.

-Response: Thank you for your precise reading of our study and sorry for unclear description of this issue which results in misunderstanding. It was clarified in revised version of manuscript. (Page 9, line 339-340; page 2, line 78-79)

3) Several abbreviations are not defined (ACME, ADE), and the same for the B/F polymorphisms. 

-Response: ACME (average causal mediation effects) and ADE (average direct effects) were defined in footnote of Table 3.

Round 2

Reviewer 1 Report

The authors have appropriately dealt with all my comments. They have also had the manuscript proofread by a native English speaker. I believe the manuscript has now improved substantially. I have only two minor comments that need addressing before publishing proceeding:

1) In the previous review, minor comment number 13, I made a typo and wrote "mmol/L" where I meant "nmol/L". Please correct this throughout the manuscript, tables and figures whenever applicable. I apologize to the authors and editors for the confusion caused.

2) In the previous review, minor comment number 14, I suggested the authors to use Means and SDs for summarising the biomarkers before and after, and the Mean Difference (MD) and 95%CIs for the change during follow-up. After reviewing the results reported by the authors in the re-analysis, and comparing them with the originals, I believe the Mean and SD, and MD and 95%CIs should be reported. As you can see, the significance as indicated by the 95%CIs (zero not included) coincides with that indicated by the p-values from the non-parametric tests (except for SICAM1 - please check). Therefore, I believe it makes more sense to report MDs as they are consistent with the means reported before and after the vitamin D supplementation. The p-values column could be omitted and the authors could write in the methods or results something along the lines of: "Despite the biomarkers change data not being normally distributed, due to the sufficiently large sample size, using non-parametric methods led to the same results as using methods that rely on the mean and SD".

Author Response

We thank the reviewers for their careful reading of the manuscript and responses. In this revision every attempt has been made to retain these positive features and improve the presentation in terms of both clarity and scientific rigor. We will now address the specific comments of each reviewer.

Reviewer # 1

1) In the previous review, minor comment number 13, I made a typo and wrote "mmol/L" where I meant "nmol/L". Please correct this throughout the manuscript, tables and figures whenever applicable. I apologize to the authors and editors for the confusion caused.

Response: No problem. It was corrected.

2) In the previous review, minor comment number 14, I suggested the authors to use Means and SDs for summarising the biomarkers before and after, and the Mean Difference (MD) and 95%CIs for the change during follow-up. After reviewing the results reported by the authors in the re-analysis, and comparing them with the originals, I believe the Mean and SD, and MD and 95%CIs should be reported. As you can see, the significance as indicated by the 95%CIs (zero not included) coincides with that indicated by the p-values from the non-parametric tests (except for SICAM1 - please check). Therefore, I believe it makes more sense to report MDs as they are consistent with the means reported before and after the vitamin D supplementation. The p-values column could be omitted and the authors could write in the methods or results something along the lines of: "Despite the biomarkers change data not being normally distributed, due to the sufficiently large sample size, using non-parametric methods led to the same results as using methods that rely on the mean and SD".

Response: Agreed and done.  (Table 2, page 7 & 8, line 267-286)

Reviewer 2 Report

The authors have made an effort to improve the manuscript. Nevertheless, the main point is that this manuscript is a description of negative results in which the effects of the Vit D intervention on the expression of the tumor biomarkers are non-significant.

If I understand well from the text what the authors find significant are the " On-study changes " (see the attached box-plots),  but this is something to be expected since they are comparing full values with their variations. The authors should clarify this point.

Minor points.

The box plot requested should be included as a Figure in the main text ( or as a Supplementary Figure).

The abreviation list should be sorted in alphabetical order. The definitions of the abbreviations I requested should be also included in the abbreviation list. There's no definition of the B/F polymorphisms. 

Author Response

We thank the reviewers for their careful reading of the manuscript and responses. In this revision every attempt has been made to retain these positive features and improve the presentation in terms of both clarity and scientific rigor. We will now address the specific comments of each reviewer.

Reviewer # 2

The authors have made an effort to improve the manuscript. Nevertheless, the main point is that this manuscript is a description of negative results in which the effects of the Vit D intervention on the expression of the tumor biomarkers are non-significant. If I understand well from the text what the authors find significant are the " On-study changes " (see the attached box-plots),  but this is something to be expected since they are comparing full values with their variations. The authors should clarify this point.

Response: Sorry for not clarifying this point which results in misunderstanding. We aimed to provide a graphical representation of pre and post values of response variable as well as their on-study changes to show just outliers. Actually, the response variables before and after vitamin D3 supplementation was compared using the Wilcoxon signed-rank test not full values with their variations. So, the on study changes columns were omitted from the attached box-plots for misunderstanding of readers. In addition, based on reviewer 1 comment, means and SDs for summarizing the biomarkers before and after, and the mean difference (MD) and 95%CIs for the change during follow-up are presented in the revised version of manuscript. Any other suggestions and comments would be most welcome. 

Minor points.

The box plot requested should be included as a Figure in the main text ( or as a Supplementary Figure).

Response: Agreed and it was added to the revised manuscript as a Supplementary Figure 2.

The abreviation list should be sorted in alphabetical order.

Response: Done

The definitions of the abbreviations I requested should be also included in the abbreviation list.

Response: It was added.

There's no definition of the B/F polymorphisms. 

Response: If I understand you correctly, you requested the definition for the VDR BsmI, FokI, ApaI, TaqI and Cdx2 polymorphisms. They were added to abbreviation list.  
